# Growth Factors, Reactive Oxygen Species, and Metformin—Promoters of the Wound Healing Process in Burns?

**DOI:** 10.3390/ijms22179512

**Published:** 2021-09-01

**Authors:** Daniela Miricescu, Silviu Constantin Badoiu, Iulia-Ioana Stanescu-Spinu, Alexandra Ripszky Totan, Constantin Stefani, Maria Greabu

**Affiliations:** 1Department of Biochemistry, Faculty of Dental Medicine, Carol Davila University of Medicine and Pharmacy, 8 Eroii Sanitari Blvd, 050474 Bucharest, Romania; daniela.miricescu@umfcd.ro (D.M.); alexandra.totan@umfcd.ro (A.R.T.); maria.greabu@umfcd.ro (M.G.); 2Department of Anatomy and Embriology, Faculty of Medicine, Carol Davila University of Medicine and Pharmacy, 8 Eroii Sanitari Blvd, 050474 Bucharest, Romania; 3Department of Plastic and Reconstructive Surgery, Life Memorial Hospital, 365 Grivitei Street, 010719 Bucharest, Romania; 4Department of Family Medicine and Clinical Base, Dr. Carol Davila Central Military Emergency University Hospital, 010825 Bucharest, Romania; constantin.stefani@umfcd.ro

**Keywords:** burns, growth factors, reactive oxygen species, metformin, wound healing, signaling pathways

## Abstract

Burns can be caused by various factors and have an increased risk of infection that can seriously delay the wound healing process. Chronic wounds caused by burns represent a major health problem. Wound healing is a complex process, orchestrated by cytokines, growth factors, prostaglandins, free radicals, clotting factors, and nitric oxide. Growth factors released during this process are involved in cell growth, proliferation, migration, and differentiation. Reactive oxygen species are released in acute and chronic burn injuries and play key roles in healing and regeneration. The main aim of this review is to present the roles of growth factors, reactive oxygen species, and metformin in the healing process of burn injuries.

## 1. Introduction

Currently, burns are the fourth most common type of injury, caused by several factors, mainly hot water, followed by hot tea or milk, hot objects, hot meals, hot oil, chemicals, electricity, sun, embers, and flame [1,2]. Depending on the degree of the damage, burns can be treated at home, by pharmacists, or may require specialist care [1]. Even though in the present there are modern healthcare services, burn-related death rates are increasing every day. Unfortunately, burn wound patients have an increased risk of cutaneous and systemic bacterial infections that can seriously delay the wound healing process [3]. Dead tissue and protein-rich exudate exist in the burn wound, providing a suitable environment for the proliferation and colonization of microbes [3]. When the skin is injured, different cell types such as keratinocytes, fibroblasts, functional cells, and growth factors (GFs) are recruited and involved in wound regeneration [4]. Both in humans and animals, a wound can be defined as damage or disruption of the normal anatomical structure and function, which can be a simple break in the epithelial integrity of the skin, or it can be deeper, affecting the subcutaneous tissue and damaging several structures, such as tendons, muscles, vessels, nerves, parenchymal organs, and bones. Burn-related wounds can be acute or chronic, according to their duration of healing [5]. Compared with acute wounds, the chronic ones have a negative impact worldwide, with huge medical costs for patients and an increased risk of mortality [6]. Chronic wounds are venous, arterial, and diabetic ulcers that are usually associated with burns, advanced age, and several systemic and blood circulation diseases [7]. Worldwide, diabetic foot ulcers represent a major health-care problem [8]. For example, in the United States of America, approximately 6.5 million people have chronic ulcers [9]. Chronic wounds can be defined as a pathologic inflammatory state, characterized by a proteolytic microenvironment at the wound site, which can lead to GFs degradation that further delays or impairs the normal wound healing process. Moreover, these types of wounds have a high bacterial load, which will delay the healing process. Clinical trials have reported that topical application of GFs on chronic wounds is for the most part unsuccessful because of their rapid degradation and extremely short half-life [10].

## 2. Wound Healing Process-General Aspects

The healing process is very complex and assumes and includes the following steps: (i) Hemostasis, (ii) inflammation, (iii) cell proliferation/granulation, and iv) the remodeling phase [11,12,13,14]. The time-frames specific for each step mentioned above in the process are 1–3 days, 3–20 days, 7–40 days, and 40 days to 2 years, respectively [13]. The normal skin wound healing is a complex process orchestrated by cytokines, GFs, clotting factors, prostaglandins, free radicals, and nitric oxide [15]. In the primary hemostasis and coagulation phase, blood platelets are present, which are small non-nucleated elements derived from megakaryocyte fragmentation exerting various essential functions [13]. Platelets present hemostatic roles such as adhesion, activation, and aggregation recruitment of other platelets and leukocytes. These functions are mediated by various specific receptors that will bind to the ligands localized sub-endothelially after vessel attrition, on other platelets, and even on leukocytes [13]. During wound healing, a coordination between epithelial cells, platelets, endothelial cells, fibroblasts, and macrophages has positive effects (Figure 1) [16].

The inflammatory phase is involved in the cleaning of the pathogens and foreign material from the wound [17]. In a condition like this, vascular permeability is increased, allowing neutrophils and monocytes to move to the wound site [17]. In the regulation of the inflammatory phase, neutrophils, which are the master regulators of this phase, will transform monocytes into macrophages [17]. Further, the macrophages will phagocytize and digest the tissue debris and the remaining neutrophils but will also secrete GFs and cytokines that will promote tissue proliferation and cell migration [17]. Corroborating these findings, the inflammatory phase is an elaborate process that involves coordination amongst a diversity of cell types. The first line of defense against tissue injury and bacterial infection is represented by neutrophils, which are granular polymorphonuclear leukocytes (PMN) [18]. Chemotaxis represents the process of migration of neutrophils and other leukocytes to infected or damaged places, accompanied by increased production of chemokines and cytokines [18]. This way, neutrophils stimulate the secretion of pro-inflammatory mediators (TNF-α, IL-1 and IL-6), anti-inflammatory cytokines (IL-8), and GFs (vascular endothelial growth factor-VEGF) that further release antimicrobial substances (cationic peptides, eicosanoids) and proteinases (elastase, cathepsin G, proteinase 3, and urokinase-type plasminogen activator) [18]. Macrophages are the key players in the transition from inflammation to proliferation. The absence of macrophages from the inflammatory or the proliferation phase of wound healing will trigger a reduced tissue formation or hemorrhage. In the early stages of wound healing, M1 macrophages are associated with phagocytic activity, stimulating the production of pro-inflammatory mediators [19]. After fulfilling the mentioned roles, M1 is converted into the M2 subset that is implicated in the synthesis of anti-inflammatory mediators and the production of extracellular matrix (ECM), in the initiation of fibroblast proliferation and angiogenic processes [19]. Moreover, M1 cells have immense phagocytic properties, swallowing the neutrophils that have suffered apoptosis and removing any pathogens or debris in the wound [20]. It has been observed that during the early stages of inflammation, around 85% of macrophages have the M1 phenotype, while 15% have the M2 phenotype. At 5–7 days post injury, this ratio is reversed when the wound is mature, where only 15–20% of macrophages have an M1 phenotype, while the majority of cells that populated the wound have the M2 phenotype. In these conditions, on 5 days post injury, M2 macrophages secrete CD301b, increased levels of anti-inflammatory cytokine (IL-10), and GFs (platelet-derived growth factor-PDGFβ and transforming growth factor-β1-TGF-β1) [20].

The inflammation phase can be defined as the primary driver of scarring at the wound site, including following burns [21]. Even though inflammation is an important step for proliferative phase stimulation, excessive inflammation may cause impairment in wound healing [22]. The proliferative phase is characterized by the accumulation of cells such as fibroblasts, keratinocytes, and endothelial cells. Proteoglycans, hyaluronic acid, collagen, and elastin—that are ECM matrix components—are implicated in the formation of the granulation tissue [11]. During this phase that may last between days and even weeks, a wide variety of cytokines and GFs, including the TGF-β family (TGF-β1, TGF-β2, and TGF-β3), the IL family, and angiogenesis factors (vascular epidermal growth factor-VEGF), are released [11]. The granulation tissue presents fibroblasts and ECM collagens. Regarding the matrix, neovascularization is sustained by angiogenesis and vasculogenesis, molecular events that will end after full epidermal coverage of the wounded surface [6]. Fibroblasts are the main agents responsible for the deposition of the new matrix. Collagen is the main component of the mature connective tissue scar, secreted by fibroblasts. Furthermore, fibroblasts secrete type III collagen in the granulation tissue, which represents the predominant form [23]. Angiogenesis is a key step in the formation of granulation tissue. Macrophages, retinal keratinocytes, and vascular smooth muscle cells are able to release VEGF, which possesses a strong ability of promoting angiogenesis [24]. Angiogenesis represents the formation of new blood vessels, an important phase of the healing process that is stimulated by basic fibroblast growth factor (bFGF), TGF-β, TNF-α, and VEGF [25]. Furthermore, neovascularization is required to support the increased proliferation of fibroblasts and keratinocytes [26]. The new blood vessels’ formation mediates the transport of nutrients, oxygen, GFs, and circulating cells, which are essential for wound healing and tissue regeneration [27]. In the final stage, the remodeling phase, type III collagen is replaced with type I collagen, and fibroblasts are differentiated into myofibroblasts, further involved in wound contraction and scar maturation [28,29].

Several cell types including the resident fibroblast, adipocytes, fibrocytes, monocytes, mesenchymal cells, and epithelial/endothelial cells can be differentiated into myofibroblasts [30]. During the normal wound healing process, myofibroblasts significantly disappear at the end of the process via apoptosis [30]. Their presence in the wound tissue is correlated with scar production. Moreover, myofibroblasts are involved in the production of ECM components and enhance the matrix elements’ synthesis, such as laminin, glycosaminoglycans, and hyaluronic acid [30]. Marconi and co-workers observed that, in gingival tissue, during the wound healing process, cytokines and GFs stimulate the differentiation of resident fibroblasts into myofibroblasts [30]. Therefore, TGF-β1 secreted mostly by M2 macrophages [20] represents the typical inducer of myofibroblast differentiation [30]. Furthermore, the keratinocyte growth factor (KGF) stimulates the release of keratinocytes that may convert fibroblasts into myofibroblasts [29]. Hesketh and colleagues observed in GW2580 mice that M2 macrophages promote scar formation. Acute wounds have an increased M2 macrophages population correlated with high levels of TGF-β1 that further induce an increased rate of fibroblast differentiation and rapid wound closure. Unfortunately, in chronic wounds, including diabetic foot ulcers and venous leg ulcers, the typical shift of M1 to M2 macrophages is dysregulated [20].

Reepithelization or the formation of the new tissue assumes the presence of the keratinocytes that migrate to the wound center from the wound edge after suffering molecular, morphological, cytoskeletal, and adhesive changes, called the epithelial–mesenchymal transition [31]. Any aberration in the remodeling phase may lead to excessive wound healing or chronic wounds [11]. In patients who suffered irradiation, delayed wound healing may be a serious clinical problem [32]. In contrast, chronic non-healing wounds are characterized by decreased production of GFs and chemokines, decreased angiogenesis, reduced proliferation and migration of several cell types such as fibroblasts, and an impaired inflammatory response [33]. In addition, diabetic chronic wounds are associated with impaired angiogenesis, leukocyte function, fibroblast proliferation, and keratinocyte migration and proliferation [34].

Extracellular vesicles (EVs) may be an excellent alternative to enhance the wound healing process, especially for diabetic chronic wounds. These vesicles may derive from mesenchymal stem cells (MSC) and contain proteins, lipids, and nucleic acids. Therefore, during skin regeneration, EVs can activate angiogenesis, proliferation, migration and differentiation of endothelial cells, fibroblasts, and keratinocytes [35].

## 3. The Impact of Growth Factors on the Healing Process and Their Clinical Applications

GFs are natural polypeptides involved in cell growth, proliferation, migration, and differentiation that are able to bind to their specific receptors located on the cell surface, further initiating signaling pathways and activating specific signaling molecules, eventually triggering protein synthesis [36,37]. In burn-related chronic wounds, GFs play a crucial role in wound healing because they may restore the physiological wound healing process [38]. In humans and animals, in the healing process, GFs such as PDGFs, FGFs, the epidermal growth factor (EGF), TGF-β, VEGF, and the granulocyte-macrophage colony stimulating factor (GM-CSF) play key roles [39,40,41,42]. In addition, GF deficiencies, including reduced levels of bFGF, PDGF, EGF, and TGF-β, have been observed in chronic pressure ulcers versus acute wounds [36]. Moreover, the expression of PDGF is more decreased in chronic dermal ulcers than in surgically created acute wounds. These aspects suggest that GF deficiencies are responsible for chronic wounds [36]. Decreased GF activity in the chronic wound may contribute to reduced amounts of ECM synthesis and collagen production [43]. ECM proteins are able to promote angiogenesis and wound healing through GFs binding. The release by the endothelial cells of the protein von Willebrand factor into plasma and in the subendothelial matrix has been shown to regulate angiogenesis [44]. Beside tissue repair, GFs are also omnipresent ingredients of colostrum, milk, and saliva, involved in digestion and nutrition [43]. GFs such as PDGF, FGF-2, IGF, and KGF, have been used in clinical trials, while PDGF is currently approved for application in human medicine [45].

### 3.1. Fibroblast Growth Factor Family and Wound Healing

FGFs belong to the growth factor family that includes 23 members, which are involved in angiogenesis, wound healing, embryonic development, and regulation of the endocrine secretion. These diverse biological roles depend on their binding to the cofactor heparin sulphate or Klotho that will form dimers with their receptors. Out of all the FGFs family, FGF-2, FGF-7 (or KGF-1), and FGF-10 (or KGF-2) have been observed to be involved in cutaneous wound healing [46]. FGFs are expressed in heart, liver, skin, and kidney tissues. Out of all FGFs members, FGF-2, -7, -10, and -21 are highly expressed in skin tissue, and all except FGF7 regulate fibroblast cell migration and activate the signaling pathway c-Jun N-terminal kinase JNK [47]. In addition, FGFs influence the expression of other GFs involved in the regenerative response, promoting organ regeneration in humans [48]. In addition, FGFs promote homeostatic balance during postnatal life [49]. FGF-1 and -2 are involved in fibroblast growth and trigger angiogenesis, cell migration, and wound healing [50].

bFGF is a member of the FGF, a highly specific mitogenic factor for many cell types, involved in wound repair, angiogenesis, nerve nutrition, and embryonic development [51]. Yang and colleagues have shown that oil bodies that express the oleosin-bFGF fusion protein have an excellent proliferative effect on NIH/3T3 cells and animal experiments, decreasing the wound size, and may accelerate the maturation of tissue granulation [51]. Moreover, FGF-2, -9, and -18 are involved in bone regeneration [52]. A positive effect of bFGF has been observed on the periodontal healing of replanted teeth [53]. bFGF is widely used for difficult wound healing treatment [54]. Recombinant human bFGF (rh-bFGF) is a multifunctional growth factor involved in cell growth and reproduction, it promotes capillary regeneration, stimulates epithelial and endothelial proliferation, and improves wound granulation tissue production. As mitogen and chemokine, rh-bFGF increases the activity and promotes the aggregation of fibroblasts [55]. In irradiated wounds, a decreased local level of bFGF in skin was observed to be associated with delayed healing. Radiation decreases bFGF production because it reduces the bFGF message in irradiated tissue. Instead, the supplemental intravenous administration of bFGF may reduce irradiated soft tissue injury [56]. Peng and co-workers observed in rats that the exogenous application of bFGF significantly upgrades the expression of VEGF and TGF-β in wound sites and enhances the angiogenesis process and cell proliferation [57]. FGF2 is involved in the proliferation and migration of keratinocytes. Instead, the effect of FGF10 (KGF2) on keratinocytes in the wound healing process appears to emanate from the connective tissue layer [58].

Ramos-Gonzalez G and colleagues conducted an experimental wound model study that included C57BL/6 mice. Human mesenchymal stromal cells seeded on collagen membranes were implanted in wounds. The control group was represented by animals with wounds without treatment or treated with collagen membranes. The study concluded that wound healing induced by mesenchymal stromal cells is associated with an increase in epithelial progenitor cells and GFs such as bFGF [59].

Xie et al. investigated the effects of bFGF on the hypertrophic scar in a rabbit ear model. The rabbit ear model of wound healing was created and treated with bFGF once daily for 3 months. The results of the study show macroscopic and histologic significant changes between scars treated with bFGF and control scars [60]. bFGF improves the quality of the wound healing process. In adults with second-degree burns, mainly with deep dermal burns, bFGF administration significantly improves the wound healing [61]. Numata et al. reported higher levels of bFGF at the wound edge of gene-knockout mice (*HDC*(+/+) especially on the third and fifth day of wound healing versus *HDC*(−/−) mice [62]. Moreover, fibrin may be a target for human bFGF, so the fibrin-binding peptide Kringle1 (K1), derived from human plasminogen, was fused to human bFGF. The recombinant K1bFGF presents increased fibrin and plasma-clot-binding ability that may induce neovascularization and enhance wound healing when applied to plasma clots [63]. Guo et al. used human-like collagen (HLC) cross-linked with transglutaminase (TG) to synthesize HLC/TG hydrogel, which was further loaded with bFGF, conducive to skin regeneration in the structure and function [64].

### 3.2. Epidermal Growth Factor and Wound Healing

EGF was the first growth factor isolated, which is involved in keratinocyte migration, fibroblast function, and granulation tissue formation [65]. Human EGF (hEGF) is a member of the GFs family that activates EGF receptors [66]. hEGF is synthesized and secreted by multiple tissues and organs of the human body. Several signaling pathways are activated by hEGF via receptors that will further promote the regulation of cell proliferation, differentiation, and migration [66]. EGF family members are constantly expressed in the tear and its receptors are expressed in ocular surface tissues [67]. EGF is recognized as an excellent wound healing factor due to its therapeutic roles, such as stimulating skin cell growth, proliferation, and differentiation [68]. In an experimental rat model, EGF in combination with Enoxparin has a synergistic effect and significantly contributes to thrombus resolution [69]. Moreover, EGF may be involved in the prevention of anastomotic leakage of esophageal atresia by accelerating the wound healing process and increasing the stability of the anastomotic line [70]. EGF might provide significant wound healing stimulation for chronic wounds compared with acute wounds [71]. The EGF receptor is widely expressed in mammals, without hematopoietic cells. EGF administration promotes migration and proliferation of both epithelial and mesenchymal skin cells [72]. Retinoic acid up-regulates the expression of EGFR in the treatment of deep partial-thickness burns using HaCaT cells [73]. A combination of EGF with curcumin bandage bioconjugate (EGF-Cur B) loaded with bone-marrow-derived mesenchymal stem cells at the wound site for diabetic wounds leads to the formation of granulation tissue, collagen deposition, and angiogenesis versus the control group [74]. EGF and FGF may regulate different key factors of wound healing in all stages. In addition, EGF and TGF-β1 synergistically stimulate cell migration, and both increase MMP-9 function and increase extracellular signal-regulated kinase ERK1/2 activation [75].

Kao et al. tested in vivo and in vitro the effects of lidocaine and human EGF-incorporated PLGA nanofibers as an anti-adhesive membrane for surgical wound healing. It is well known that hEGF facilitates dermal regeneration by promoting the stimulation, proliferation, and migration of keratinocytes, endothelial cells, and fibroblasts. The study concluded that the combination used offers post-operative pain relief and wound healing [76].

In the treatment of chronic wounds, EGF conjugated with hyaluronate (HA) enhances the healing effect [77]. This aspect is sustained by the long-term stability against enzymatic degradation. The therapeutic effect is manifested by the higher biological interaction with HA receptors on skin cells [77]. Moreover, exogenous EGF treatment increases the rate of wound healing in a three-dimensional stem cell-derived model of vocal fold mucosa [78]. Kaya and co-workers tested the effect of EGF on rats after an early burn. The animals were divided in three groups: Group 1 (no post-burn treatment), group 2 (treated with physiological saline solution), and group 3 (EGF injected intra-dermally into the stasis zone). The highest surviving rate was observed in group 3 (70.2%) compared with the others two group. No difference was detected between groups 1 and 2 [79]. EGF protects the zone of stasis tissue from burn damage [79]. It can also correct the pseudo-healing of traumatic tympanic membrane perforations, after daily application over six months, to keep the membrane moist [80]. In patients with diabetic foot ulcers, parenteral administration of EGF on epithelial tissue has a positive effect on the healing process [81]. Dogan and colleagues conducted an experimental study performed on male Wistar-Albino rats divided into non-diabetic and diabetic rats and reported that EGF loaded in a gelatine-microsphere improves wound healing both in normal and diabetic rats [82]. Wei et al. tested the effect of a recombinant (rhEGF) using HaCaT and L929 cells [83]. In vitro, a concentration of 10 ng/mL of rhEGF promotes the proliferation and migration of epithelial cells and fibroblasts. On the other hand, in vivo, vacuum sealing drainage combined with rhEGF, can improve wound healing better than the other treatments [83]. Moreover, the fabrication of scaffolds such as poly (ε-caprolactone) (PCL) containing EGF has high potential for wound-healing applications [84]. Kalay and co-workers reported that EGF may act like an antioxidant in wound tissue. Scavenging the wound against toxic compounds may contribute to the healing in earlier stages, becoming a possible antioxidant used in therapies [85].

### 3.3. Transforming Growth Factor-Beta Family and Wound Healing

The TGF-β superfamily includes three isoforms, TGF-β1, -β2, and -β3, coded by genes with different locations in various chromosomes [86]. TGF-β are soluble extracellular proteins, which regulate the development in both vertebrates and invertebrates [87]. TGF-α belongs to the EGF family; it is a 50-amino-acid polypeptide, which in humans has 10 members, isolated from a conditioned medium of virally transformed cells and tumor cells [88]. TGF-β members regulate cell functions such as migration, apoptosis, proliferation, and differentiation [87]. TGF-β is secreted in a latent form by most cells and is activated via TGF-β receptor signaling [89]. TGF-β also plays a major role in myofibroblast transdifferentiation and has been involved in numerous fibrotic eye diseases [90]. TGF-β1 is found in the granulation tissue of diabetic foot ulcers [11]. This cytokine accelerates wound healing [91] because it is a key regulator of the mesenchymal cells, involved in the production and remodeling of the ECM [92]. TGF-β inhibits tissue protease production and stimulates the secretion of the inhibitors of matrix metalloproteinases (MMPs) [93]. Using animal models, the expression of endogenous TGB-β1 is reduced in impaired wound healing, while its exogenous addition may improve healing [92,94]. Peplow and co-workers observed in cutaneous wounds of animals with diabetes that TGF-β has decreased gene expression and content in early phases of healing for diabetic wounds versus the nondiabetic wounds [95]. In horses, elevated levels of TGF-β may result in the production of exuberant granulation tissue [96]. Rorison and co-workers reported that in patients with excellent post-burn healing, in the first two weeks post-injury, plasma levels of TGF-β1 rapidly increase to significantly higher levels and then rapidly decrease. However, in patients who develop hypertrophic scarring, plasma levels of TGF-β levels in the early stages after a burn are absent [97].

### 3.4. Vascular Endothelial Growth Factor Family and Wound Healing

The VEGF family is a group of homodimeric proteins that contain six members: VEGF-A, VEGF-B, VEGF-C, VEGF-D, VEGF-E, and the Placental growth factor (PlGF) [98]. VEGF is the most abundant form, a specific mitogen for endothelial cells, involved in the proliferation, migration, and activation of endothelial cells [98,99]. Moreover, VEGF promotes permeability and fenestration of blood vessels. VEGF-C and -D are important for lymphangiogenesis, while VEGF-B has a role in embryonic angiogenesis. PlGF is involved in pathological angiogenesis [98]. Beside endothelial cells, many cell types involved in the wound healing mechanism possess VEGF receptors, such as keratinocytes, mesenchymal stem cells [100], fibroblasts, smooth muscle cells, platelets, neutrophils, and macrophages [101]. Humans have two types of VEGF receptors, Flt-1 (VEGFR-1) and KDR (VEGFR-2), both with high affinity and members of the type III tyrosine kinase family [101]. In the skin, VEGF is produced by keratinocytes, macrophages, platelets, and tumor cells [100]. During granulation tissue formation, capillaries and lymphatic vessels are formed from the pre-existing vessels at the wound site with the help of FGF and VEGF [102,103,104]. VEGF can induce healing by aiding in vascular permeability and prevents the inflammatory cells from migrating to the injured place and accelerates the proliferation and migration of endothelial cells [102]. Lord et al. conducted an experimental study on animals and observed that plasmid DNA encoding perlecan domain I and VEGF189-loaded scaffolds promote dermal wound healing in normal and diabetic rats. The treatment administered to rats increased the number of blood vessels and sub-epithelial connective tissue matrix components [105]. Shi and co-workers performed an experimental study using rat epidermal stem cells ESCs that have been treated with exogenous VEGF in order to explore the healing of deep partial-thickness burn wounds. The study concluded that VEGF can promote the healing of deep partial-thickness burn wounds in rats [106]. The clinical study conducted by Li et al. that included children with femoral fracture after surgery explores the effects of VEGF incorporated into silver nanoparticles (Ag NPs) on the wound healing process. The results reported that Ag NPs loaded with VEGF improve the femoral fracture healing and especially the blood vessel repair [107].

Vijayan et al. prepared a heparin-based Chitosan-Polyethylene glycol (CS-PEG-H) scaffold, which was able to deliver GFs, such as VEGF and bFGF, in a controlled manner to the wound site. The study revealed an elevated wound healing rate, including neovascularization versus the control on day 10 [108]. Saaristo and co-workers conducted an experimental study on genetically diabetic db/db mice to explore the effects of VEGF-C and the role of endogenous VEGF-C and VEGF-D during tissue repair. The results of the study concluded that VEGF-C enhances the wound healing process and attests to its applicability in the treatment of complicated diabetic wounds [109]. LDL, but not oxLDL, enhances the expression of VEGF. Bogachkov and co-workers observed in an experimental study that hypercholesterolemic apolipoprotein E-deficient (*ApoE*^−/−^) mice display a delayed wound healing process compared to age-matched C57/BL6 wild-type controls after skin punch biopsy [110].

### 3.5. Insulin-Like Growth Factor Family and Wound Healing

The IGF family is composed of two peptide ligands, IGF-1 and IGF-2, and the hormone insulin [111]. IGF-1 is produced mainly by the liver under the control of the growth hormone, being involved in body growth. Apart from its role in growth, IGF-1 plays key roles in maintaining the normal function of kidneys, cardiovascular system, and brain [112]. Moreover, IGF plays an important role in cartilage homeostasis, maintaining a balance between proteoglycan synthesis and breakdown [113]. Regarding IGF-1 and wound healing, the growth factor has multiple functions, having a chemotactic effect on endothelial cells, stimulating the proliferation of fibroblasts and keratinocytes, and enhancing the wound strength. Botusan and co-workers conducted a study to observe the role of circulating IGF-I on the wound healing rate. The study included mice with deficiency of liver-derived IGF-I (LI-IGF-I-/- mice) divided in two groups, the normoglycemic—the control—group and the second one composed of diabetic animals and concluded that IGF-I does not affect wound healing in the two mice groups [112]. In contrast, Gong and co-workers explored the effects of exogenous IGF-1 on the healing process of skin ulcers in diabetic male rats. The experimental study animals were divided into the control group, the model group, and the IGF-1 treatment group, which received different doses of IGF-1. The study results reported that IGF-1 promotes the wound healing of skin ulcers in diabetic rats [114]. Shinchi et al. revealed in a rabbit model that IGF-1 may sustain the collagen release and significantly improves the wound healing and may prevent the urethral stricture after urethral injury [115]. IGF-1 has a therapeutic effect even in the healing of gingival mucosa damage [116]. Using rats, Todorović et al. evaluated the expression of IGF-1 in skin cells and its systemic and cutaneous tissue concentrations during the acute phase of wound healing. The study results support a paracrine role of IGF-I during the acute phase of a wound [117].

Reckenbeil and colleagues investigated the effects of IGF1 on proliferation, wound healing, and differentiation processes on the human periodontal ligament. The study concluded that an improvement in wound healing and proliferation processes was observed, as well as sustained cell differentiation under inflammatory stimuli [118]. Additionally, Balaji and co-workers demonstrated, using two different models (in vivo and in vitro), that IGF-1 enhances the wound healing process and induces angiogenesis via a VEGF-independent pathway in chronic wounds [119]. Using Wistar adult male rats, divided into diabetic and non-diabetic, it has been revealed that topical administrations of 1% and 3% IGF-1 creams enhance the expression of myofibroblasts in the rats’ wound-healing process [120].

### 3.6. Keratinocyte Growth Factor and Wound Healing

In an injury, keratinocytes are involved in the reepithelization process, interacting with dermal cells and collagen-rich ECM to re-establish the coverage of the wound bed [121]. KGF stimulates the multiplication and migration of keratinocytes [122] and induces cell proliferation and motility [123]. Devalliere et al. investigated in vivo the healing effect of KGF and the cellular protective peptide (ARA290) fused with the elastin-like peptide (ELP). The study was performed on a diabetic wound model and tested the effect of ARA290-ELP and KGF-ELP alone or in combination, and it revealed that ARA290-ELP was able to accelerate healing by increasing angiogenesis. Both ARA290-ELP and KGF-ELP are promising new therapeutics for the treatment of chronic wounds [124]. Bienert et al. evaluated the capacity of FGF-, EGF-, KGF-, PDGF-, or VEGF to promote wound healing in vitro. The study confirms their potential as wound-healing therapies [125]. Chomiski and colleagues tested the impact of KGF treatment on the expression of wound-healing-related genes using keratinocyte cells culture obtained from burn patients. The study included keratinocyte cells divided into four groups, as follows: TKB (KGF-treated keratinocytes from burn patients), UKB (untreated keratinocytes from burn patients), TKC (KGF-treated keratinocytes from controls), and UKC (untreated keratinocytes from controls) and concluded that there is no difference between wound-healing-related genes expression [126]. Moreover, in corneal injury, KGF-2 manifests better effects in re-epithelialization, acceleration of migration, and reduction of scar formation, compared with bFGF [127]. Hydroxyethyl-methacrylate (HEMA) copolymers labelled with KGF increase the release to the wound interface during the first 20 min of the treatment and enhance the potential of re-epithelialization [128].

### 3.7. Granulocyte-Macrophage Colony Stimulating Factor and Wound Healing

GM-CSF, which is the founding member of the β common cytokine family, was identified for the first time in the context of hematopoiesis. GM-CS together with the other two family members, IL-3 and IL-5, are pleiotropic regulators of inflammation in the response to pathogens. Moreover, these cytokines are involved in autoimmune diseases and cancer progression [129]. GM-CSF plays an essential role in the wound healing cascade, being involved in the stimulation and recruitment of keratinocytes, macrophages, and fibroblasts [130] and it regulates the inflammatory response [131]. Controlled trials have shown that GM-CSF is able to accelerate the healing of chronic wounds [130]. GM-CSF is a pleiotropic cytokine that can activate granulocyte and macrophage cell lineages, even involved in wound healing of human corneal epithelial cells [132]. The recombinant human granulocyte macrophage colony stimulating factor (rhGM-CSF) presents a good effect on healing in both deep burn wounds and leprosy ulcers. The healing potential of rhGM-CSF in second-degree burns is very important, because these burns represent a dilemma regarding the medical intervention; sometimes the surgery may be carried out rapidly, other times it should be delayed until remnant dermal components are re-epithelialized [133]. Li and colleagues published the results of a systematic review and meta-analysis and supported the potential of rhGM-CSF to be safe for the treatment of the deep second-degree burns [134]. Chi et al. tested the effect of rhGM-CSF for healing burns on children. The subjects were randomly divided into two groups, an experimental group that received external rhGM-CSF gel and a control group who received rhGM-CSF gel matrix components. After the application on the burn surface of rhGM-CSF, the healing time was shorter, without significant adverse reactions [135]. Sun and co-workers assessed the effects of rhGM-CSF gel in patients with full-thickness frostbite wounds on the foot and hand. Topical administration of the rhGM-CSF gel has beneficial effects, shortening the time of wound healing and reducing the inflammatory response of the wound [136]. Moreover, Lim et al. explored the effect of GM-CSF on vocal fold wound healing, both in vivo and in vitro, and confirmed its therapeutic potential promoting regeneration [137]. Exogenous administration of GM-CSF in diabetic mice enhanced wound healing, associated with elevated levels of IL-6 and monocyte chemoattractant protein-1 that are correlated with increased neovascularization, and infiltration of macrophages and neutrophils. By contrast, GM-CSF did not present any beneficial effects in nondiabetic wound healing [138].

### 3.8. Platelet-Derived Growth Factor Family and Wound Healing

The PDGF family contains homo and heterodimeric GFs such as PDGF-AA, PDGF-AB, PDGF-BB, PDGF-CC, and PDGF-DD, exerting their functions after binding to transmembrane tyrosine kinase receptors. PDGF has chemotactic properties on neutrophils, monocytes, and fibroblasts. Moreover, it enhances fibroblasts proliferation, their contract to collagen matrices, ECM production, and finally induces the myofibroblast phenotype in these cells [139]. In wound fluid, PDGF is released from degranulating platelets. Studies performed in vivo have shown that PDGF increases capillaries’ structural integrity by the recruitment of pericytes and smooth cells. During the epithelization phase of wound healing, PDGF is involved in the up-regulation of IGF-1 and thrombospondin-1 [140]. During tissue repair, various cell types are able to express PDGF-AA, having a mitogen effect on mesenchymal cells that express the receptor PDGF-Rα. After binding to the receptor PDGF-Rα, PDGF will play an important role in the granulation tissue formation and myofibroblast differentiation/maturation. The expression of PDGF-AA is reduced in mice with impaired wound healing [141].

## 4. Oxygen Metabolism and Wound Healing Process

For normal wound healing, a proper oxygen supply is needed, which is involved in cell proliferation, migration, and angiogenesis, enhancement of collagen deposition, and epithelization of regenerative tissue [142]. Mitochondria, endoplasmic reticulum (ER), peroxisomes, several oxidase enzymes, and phospholipid metabolism can generate reactive oxygen species (ROS), both free radicals and nonradical species. Hydrogen peroxide (H_2_O_2_) is the main ROS produced in wounds by NADPH oxidase (NOX), which acts as a chemotactic signal in the first minutes after wounding [143]. Beside H_2_O_2_, NOX can generate anion superoxide ($O_{2}^{•\_})$, both having anti-microbial properties, and may be able to destroy the surrounding tissue and cells [144,145]. Neutrophils have a chemotactic response to exogenous H_2_O_2_ [146]. During the inflammatory phase of the chronic wounds, excessive neutrophils and macrophages generate elevated levels of ROS [147]. Likewise, during the γ) and pro-inflammatory cytokines (IL-1, IL-12, TNF-α) produce $O_{2}^{•\_}$ and H_2_O_2_ [148]. On the other hand, H_2_O_2_ may increase the expression of mRNA for IL-6, IL-8, and TNF-*α* [149]. Moreover, $O_{2}^{•\_}\;$ released by neutrophils in chronic wounds has been shown to be increased by 170%, compared to acute wounds [147]. ROS generated in wounds play a key role in healing and regeneration [150]. During the respiratory burst, the isoform NOX2 plays a predominant role in the anti-microbial fight, being responsible for the production of huge amounts of ROS. Thus, inside phagocytes, elevated concentrations of ROS will have a toxic effect for phagocytosed microbes by DNA damage, lipid peroxidation, and amino acid oxidation [151]. Redox signaling is important for cellular renewal, migration, and proliferation, while an excess of ROS perturbates tissue homeostasis [152]. Oxidative stress (OS) affects cell growth, while in the absence of reactive nitrogen species (RNS), ROS exert a slight positive effect on cell migration and wound healing that is not associated with cellular morphology change [153]. Injectable ROS have the potential of healing, while oxidative damage induced by ROS has a negative effect [154]. Currently, hyperbaric oxygen therapy is used in the medical field in a variety of disorders, including radiation-induced tissue injuries, non-healing states associated with ischemia, and malignant neoplasms [143]. By contrast, decreased levels of ROS such as $O_{2}^{•\_}$ and H_2_O_2_ have growth factor-like properties that may induce DNA synthesis in quiescent cells and expression of c-fos and c-myc genes associated with proliferation [155]. H_2_O_2_ is able to activate several signaling pathways, including phosphatases and protein kinases, and even transcription factors [156].

## 5. Growth Factors, Reactive Oxygen Species and Signaling Pathways Involved in Wound Healing

GFs may activate the phosphatidylinositol 3-kinase/protein kinase B (PI3K/AKT) pathway by phosphorylation, promoting angiogenesis. This molecular pathway has important cellular functions including cell adhesion, migration, and angiogenesis [157]. VEGF activates PI3K/AKT and the mitogen-activated protein kinases (MAPK) signaling pathways [158,159]. MAPK is activated by keratinocytes, predominant cells in the epidermis [160]. In many types of cells and tissues, the MAPK signaling pathway is involved in inflammation, cell proliferation, and differentiation [161]. The MAPK family are serine/threonine kinases, including three subfamilies, extracellular signal-regulated kinases 1 and 2 (ERK1/2), P38, and JNK/SAPK, mediating a variety of cell functions, such as proliferation (ERK1/2) and migration [162]. ERK1/2, JNK, and p38 MAPK have the following activation sites: Thr202/Tyr204, Thr183/Tyr185, and Thr180/Tyr182, respectively [163]. The expression of TNF-α, which is necessary for the initiation of wound healing process, depends on the activation of MAPK, p38, JNK, ERK, and on the nuclear factor (NF) κB pathway [164]. It has been shown in vitro that JNK was able to induce keratinocyte migration and wound repair, while its inhibition suppressed cell proliferation at the wound site, and further delayed wound closure [165]. The EGF receptor (EGFR) located on the cell membrane stimulates the MAPK signaling pathway, which in turn initiates various biological processes, such as the proliferation/differentiation of cells, tissue remodeling, and wound repair [166]. ERK plays a central role in the regulation of cell migration, proliferation, differentiation, and cell survival [167]. While the precise role of the p38 MAPK signaling pathway during wound healing has not yet been fully elucidated, recent studies have suggested its involvement in keratinocytes’ migration [167]. When the NF-κB pathway is activated by inflammatory stimuli, it migrates to the cell nucleus, binds with DNA molecules, and induces the transcription of a wide variety of genes involved in inflammation, proliferation, migration, cell cycling, and inhibition of apoptosis [167]. An inappropriate activation of MAPK, especially of JNK, is considered to have a critical role in insulin resistance development [168]. ROS and RNS can modulate all three members of the MAPK family signaling pathway [169] and PI3K/AKT [170]. FGF receptors phosphorylate specific Tyr residues that mediate the interaction with cytosolic adaptor proteins of the signaling pathways MAPK and PI3K/AKT [171].

## 6. Metformin and Wound Healing

Metformin is a biguanide, considered the most used hypoglycemiant drug in type 2 diabetes mellitus [172], being the first-line therapeutic agent in diabetic patients [173]. In severe burns, metformin is used for the long-term control of glycemia [174]. It is important to know that, in severely burned patients, metformin is as effective as insulin in lowering the serum glucose levels and rarely produces hypoglycemia [174]. It acts at multiple levels on the glucose metabolism (reducing hepatic gluconeogenesis, increasing insulin sensitivity, enhancing peripheric glucose uptake, increasing nonoxidative glucose metabolism) [175], lipid metabolism (augmenting the oxidation of fatty acids in liver and decreasing lipogenesis) [175], protein metabolism (increasing the rate of protein synthesis) [176], energetic metabolism [177], and has influence upon inflammation (mainly anti-inflammatory) [178,179,180] and mitophagy (increasing mitophagy) [181,182]. We must consider the influence of metformin upon wound healing in two ways: Indirectly, by acting upon the metabolisms and inflammation, and directly, by acting upon the scarring and epithelization processes.

a. The indirect effect. Metformin is one of the main drugs used for the pharmacological modulation of the hypermetabolic state and the systemic inflammatory response, which characterizes severe burns [174]. Metformin administration results in controlling hyperglycemia comparable to insulin administration, but with a lower risk for iatrogenic hypoglycemia [174]. Metformin reduces lipogenesis [175] and increases the rate of protein synthesis [176,183]. It has been proved that metformin administration decreases insulin resistance in liver, fat, and skeletal muscle with overall reduction of the hypercatabolic state of severely burned patients [184,185,186]. Better control of the prolonged hypercatabolic state in severe burns accelerates the wound healing, diminishes the rate of infection, and improves the survival [174,187].

By increasing the protein synthesis rate [174,181], decreasing insulin resistance [175,176], and controlling hyperglycemia [174], metformin creates the metabolic premises for better and faster healing of the wounds and facilitates the skin graft take. These effects were proven in clinical trials in burned patients [172,185] and in experimental studies in diabetic rats [188]. Many burns surgeons and intensive burn care doctors have been noticing that a mitigation of the hyperglycemic-hypercatabolic state of the severe burned patient is followed by more rapid wound healing and improved survival rates [187]. In experimental models, it was revealed that metformin improved skin flap survival through the Nitric Oxide (NO) system [189].

Experimental studies on animal models recently demonstrated that metformin modulates the adenosine monophosphate-activated protein kinase (AMPK)–mammalian target of rapamycin (mTOR)–transcription factor EB (TFEB) pathway [190] (Figure 2), which is known to regulate the autophagy process [191].

The activation of autophagy results in the inhibition of apoptosis, stimulation of angiogenesis, and reduction of oxidative stress, with consequent better survival of random skin flaps [190].

b. The direct effect. There are studies concerning the direct effect of metformin upon the healing process of wounds. In animal models, it appears that metformin inhibits the wound healing; in human patients with diabetes mellitus and foot ulcers, metformin increases the ulcer size [192]. These effects are explained by the fact that metformin reduces cell (keratinocytes) proliferation, by interfering at least with AMPK and mTOR [193,194], resulting in altering the cell cycle of keratinocytes, without inducing apoptosis [192].

Clinical studies on pediatric burned patients revealed that metformin inhibits the epithelial–mesenchymal transition (EMT) in fibroblasts isolated from burned patients [195]. Practically, metformin downregulates genes related to fibrosis and the epithelial–mesenchymal transition [195]. Furthermore, metformin mitigates the proliferation of fibroblasts and of keratinocytes [192,195]. This type of metformin action might not be beneficial for the burned wound healing per se but might be useful for preventing hypertrophic scarring post-burns. Further studies need to be developed in order to quantify the effects of metformin on wound healing in burned patients.

## 7. Conclusions

The wound healing process involves the following steps: Hemostasis, inflammation, cell proliferation, and tissue remodeling, with platelets, neutrophils, monocytes, macrophages, fibroblasts, keratinocytes, and myofibroblasts all participating. GFs are natural polypeptides present in a higher concentration in acute burn injuries, while in chronic ones, their concentration is more decreased. ROS are produced mainly by the NADPH family, and H_2_O_2_ and $O_{2}^{•-}$ are especially elevated in chronic wounds versus the acute form and are implicated in the anti-microbial fight and further in healing. Several GFs are able to activate the PI3K/AKT/mTOR or MAPK signaling pathway, promoting cell growth, migration, proliferation, and survival.

## Figures and Tables

**Figure 1 ijms-22-09512-f001:**
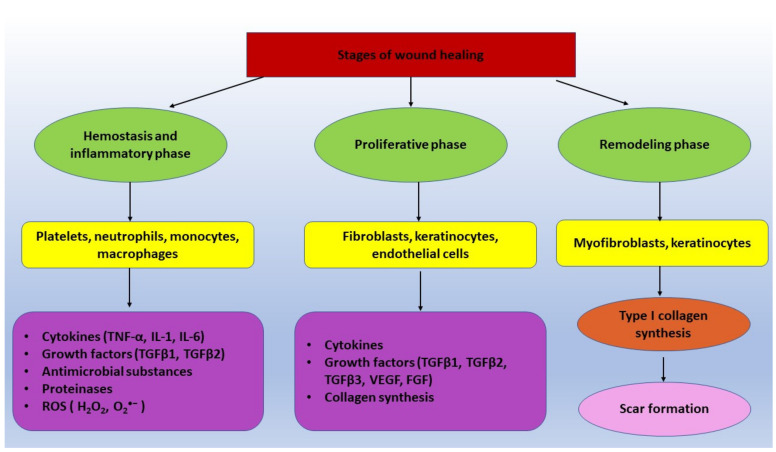
Stages of wound healing.

**Figure 2 ijms-22-09512-f002:**
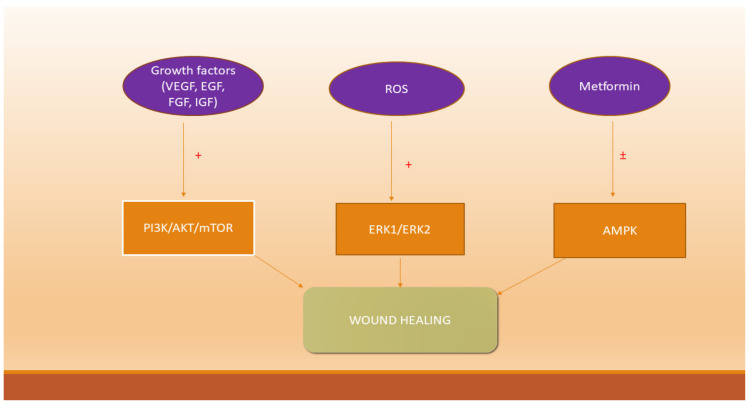
Growth factors, reactive oxygen species, metformin, and signaling pathways involved in the wound healing process (+ activation, − inhibition).

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
