# Peer review of "Growth Factors, Reactive Oxygen Species, and Metformin—Promoters of the Wound Healing Process in Burns?"

_ijms, 2021, doi:10.3390/ijms22179512_

Round 1
Reviewer 1 Report
General comments
I would like to thank the authors for this interesting manuscript titled “Growth factors, reactive oxygen species and metformin- promoters of the wound healing process in burns?”. The paper has sufficiently utilized relevant sources to review the chosen topic. Moreover, I find the topic exciting and important. However, I also did notice minor errors throughout the paper that require editing.
1.Introduction
Line 35: there is double space behind dot, you should change to single space.
Line 42: there is double space behind dot, you should change to single space.
- Wound healing process- general aspects
Line 61: you should abbreviate the word “growth factors” (GFs), you already did that on line 38.
Line 83: I recommend you first enunciate the word “vascular endothelial growth factor” and then abbreviate it.
Line 93: there is double space behind dot, you should change to single space.
Line 100: there is double space behind comma, you should change to single space.
Line 101: you should correct the space before the word inside the parenthesis, and you should abbreviate the word “growth factors” (GFs), you already did that on line 38.
Lines 110-136: you should change the font size from 9 to 10.
Line 110: there is double space behind words “that” and “between”, you should change to single space.
Line 112: you should correct the space before the word inside the parenthesis.
Line 120: there is double space behind dot, you should change to single space.
Line 121: I recommend you first enunciate the word “basic fibroblast growth factor” and then abbreviate it.
Line 124: you should abbreviate the word “growth factors” (GFs), you already did that on line 38, and you should correct double space behind comma.
- The impact of growth factors on the healing process and their clinical applications
Lines 144-145: you can abbreviate the words “platelet derived growth factors”, “transforming growth factor beta” and “vascular endothelial growth factor” without stating them before, you had already abbreviated them before in the text.
Line 159: you should not abbreviate terms in titles.
Line 162: there is double space behind dot, you should change to single space.
Line 171: there is double space behind dot, you should change to single space.
Line 173: you can abbreviate the term “basic fibroblast growth factor”, you had already abbreviated them in line 121.
Line 175: there is double space behind dot, you should change to single space. In the same line, you should write “Yang and colleagues”, without the "J", and you should change the font size from 9 to 10.
Line 186: there is double space behind dot, you should change to single space.
Line 188: there is double space behind word “Peng”, you should change to single space. Moreover, you should write “Peng and co-workers”, without the "J", and you should change the font size from 9 to 10.
Line 194: you should write “Ramos-González and colleagues”, without the "G", and you should change the font size from 9 to 10.
Line 199: you should abbreviate the word “growth factors” (GFs) you already did that on line 38.
Line 200: you should write “Xie et al.”, without the "JL" and without italics.
Line 203: there is double space behind dot, you should change to single space.
Line 206: you should write “Numata et al.”, without the "Y" and you should change the font size from 9 to 10.
Line 207: there is double space behind word “versus”, you should change to single space.
Line 211-212: you should write “Guo et al.”, without the "Y" and without italics. Moreover, you should change the font size from 9 to 10. On the other hand, there is double space behind word “Guo”, you should change to single space.
Line 213: there is double space behind word “with”, you should change to single space.
Line 215: you should not abbreviate terms in titles.
Line 224: there is double space behind dot, you should change to single space.
Line 236: there is double space behind dot, you should change to single space.
Line 240: you should write “Kao et al.”, without the "CW".
Line 251: you should write “Kaya and co-workers”, without the "O".
Line 255: there is double space behind dot, you should change to single space.
Line 259: you should write “Doga and coleagues”, without the "S" and change the font size from 9 to 10.
Line 262: you should write “Wei et al.”, without the "S".
Line 268: you should write “Kalay and co-workers”, without the "Z" and change the font size from 9 to 10.
Line 271: you should not abbreviate terms in titles.
Lines 286-287: you should write “Peplow and co-workers”, without "PV" and change the font size from 9 to 10.
Line 289: choose between TGF-beta or TGF-β and use the same term throughout the text.
Line 290: you should write “Rorison and co-workers”, without "P" and change the font size from 9 to 10. Moreover, there is double space behind dot, you should change to single space.
Line 293: there is double space behind dot, you should change to single space.
Line 295: you should not abbreviate terms in titles.
Line 296: remove the VEGF term in parentheses.
Line 309: there is double space behind dot, you should change to single space.
Line 312: you should write “Lord et al.”, without "MS" and without italics. Moreover, you should change the font size from 9 to 10.
Line 316: you should write “Shi and co-workers”, without "Y". Moreover, you should change the font size from 9 to 10.
Line 320: you should write “Li et al.”, without "X" and without italics. Moreover, you should change the font size from 9 to 10.
Line 324: you should write “Vijayan et al.”, without "A" and without italics.
Line 325: you should abbreviate the word “growth factors” (GFs) you already did that on line 38.
Line 327: you should write “Saaristo.”, without "A".
Line 332: you should write “Bogachkov and co-workers”, without "YY”.
Line 336: you should not abbreviate terms in titles.
Line 338: remove dot before the bibliographic citation.
Line 345: you should write “Botusan and co-workers”, without "IR”.
Line 349: there is double space behind dot and behind comma, you should change to single space. Moreover, you should write “Gong”, without "F” and you should change the font size from 9 to 10.
Line 354: you should write “Sinchi et al.”, without "M” and without italics. Moreover, you should change the font size from 9 to 10.
Line 357: there is double space behind dot, you should change to single space. Moreover, you should write “Todorovic et al.”, without "V” and without italics. Finally, you should change the font size from 9 to 10.
Line 359: there is double space behind dot, you should change to single space.
Line 361: you should write “Reckenbeil and colleagues.”, without "J” and without italics. Moreover, you should change the font size from 9 to 10.
Line 365: you should write “Balaji and co-workers”, without "S”.
Line 370: you should not abbreviate terms in titles.
Line 374: you should write “Devalliere et al.”, without "J” and without italics. Moreover, you should change the font size from 9 to 10.
Line 379: you should write “Bienert et al.”, without "M” and without italics.
Line 381: there is double space behind dot, you should change to single space.
Line 381-382: you should write “Chomiski and colleagues.”, without "V”.
Line 393: you should not abbreviate terms in titles.
Line 396: you should remove parenthesis of IL-3.
Line 409: you should write “Li.”, without "J". Moreover, you should change the font size from 9 to 10.
Line 411: you should write “Chi et al.”, without "YF” and you should change the font size from 9 to 10. Moreover, there is double space behind word “of”, you should change to single space.
Line 415: you should write “Sun and co-workers”, without "ZA”. Moreover, you should change the font size from 9 to 10.
Line 419: you should write “Lim et al.”, without "JY” and without italics.
Line 425: you should not abbreviate terms in titles.
Line 431: there is double space behind dot, you should change to single space.
- Oxygen metabolism and wound healing process
Line 450: you should remove space before comma.
Line 459: there is double space behind dot, you should change to single space.
Line 461: there is double space behind dot, you should change to single space.
Line 464: there is double space behind dot, you should change to single space.
- Growth factors, reactive oxygen species and signalling pathways involved in wound healing
Line 481: there is double space behind dot, you should change to single space.
Line 491: you should change the font size from 9 to 10.
Line 492: there is double space behind dot, you should change to single space.
Line 499: there is double space after word “cell” and after comma.
- Metformin and wound healing
Line 510: there is no space behind the dot, you should correct it.
Line 516: there is no space before parenthesis, you should correct it.
Lines 521-530: you should change the bold font to normal format and justify the paragraph.
Line 528: there is double space behind dot, you should change to single space.
Line 535: there is no space before parenthesis, you should correct it.
Lines 535-536: I recommend you first enunciate the words “Adenosine Mono Phosphate Kinase” and “Mammalian Target of Rapamycin” and then abbreviate it.
Conclusions
Line 552: there is double space behind dot, you should change to single space.
Lines 558-559: you should put a space between the Conclusions section and the Author Contributions section.
I would like to suggest to the authors to correct the minor errors in the paper in order to resubmit it, and one more time, I would like to congratulate to the authors on submitting your review article.
I wish the authors the best of luck in their editing process.

Author Response
In attach, cover letter

Reviewer 2 Report
The present review paper is well written and organized. The chosen topic is very interesting.
Authors should add a sub paragraph on the role of extracellular vesicles derived from stem cells and their potential role in the regenerative medicine.
They also should deepen the role of myofibroblasts in the wound healing process.
(for e.g. doi: 10.3389/fphys.2021.676512).
Author Response
In attach, cover letter

Round 2
Reviewer 2 Report
Authors have been reply to all my comments.